# Role of Cholesterol in the Regulation of Hydrogen Sulfide Signaling within the Vascular Endothelium

**DOI:** 10.3390/antiox11091680

**Published:** 2022-08-28

**Authors:** Perenkita J. Mendiola, Emily E. Morin, Laura V. Gonzalez Bosc, Jay S. Naik, Nancy L. Kanagy

**Affiliations:** 1Department of Physiology, Medical College of Georgia, Augusta Univeristy, Augusta, GA 30912, USA; 2Department of Cell Biology and Physiology, University of New Mexico Health Science Center, Albuquerque, NM 87131, USA

**Keywords:** microdomains, endothelial cell, methyl-β-cyclodextrin

## Abstract

H_2_S is a gaseous signaling molecule enzymatically produced in mammals and H_2_S-producing enzymes are expressed throughout the vascular wall. We previously reported that H_2_S-induced vasodilation is mediated through transient receptor potential cation channel subfamily V member 4 (TRPV4) and large conductance (BK_Ca_) potassium channels; however, regulators of this pathway have not been defined. Previous reports have shown that membrane cholesterol limits activity of TRPV4 and BK_Ca_ potassium channels. The current study examined the ability of endothelial cell (EC) plasma membrane (PM) cholesterol to regulate H_2_S-induced vasodilation. We hypothesized that EC PM cholesterol hinders H_2_S-mediated vasodilation in large mesenteric arteries. In pressurized, U46619 pre-constricted mesenteric arteries, decreasing EC PM cholesterol in large arteries using methyl-β-cyclodextrin (MBCD, 100 µM) increased H_2_S-induced dilation (NaHS 10, 100 µM) but MBCD treatment had no effect in small arteries. *Enface* fluorescence showed EC PM cholesterol content is higher in large mesenteric arteries than in smaller arteries. The NaHS-induced vasodilation following MBCD treatment in large arteries was blocked by TRPV4 and BK_Ca_ channel inhibitors (GSK219384A, 300 nM and iberiotoxin, 100 nM, respectively). Immunohistochemistry of mesenteric artery cross-sections show that TRPV4 and BK_Ca_ are both present in EC of large and small arteries. Cholesterol supplementation into EC PM of small arteries abolished NaHS-induced vasodilation but the cholesterol enantiomer, epicholesterol, had no effect. Proximity ligation assay studies did not show a correlation between EC PM cholesterol content and the association of TRPV4 and BK. Collectively, these results demonstrate that EC PM cholesterol limits H_2_S-induced vasodilation through effects on EC TRPV4 and BK_Ca_ channels.

## 1. Introduction

Vascular ECs play an important role in maintaining vascular homeostasis by producing and balancing vasodilatory, coagulative, proliferative, and inflammatory factors. H_2_S has been shown to contribute to endothelial health by activating these pathways [1,2,3,4,5,6,7,8,9,10], demonstrating the importance of H_2_S to maintain a healthy vasculature and paving the way for the development of H_2_S-releasing compounds entering clinical trials [3,5]. This exciting breakthrough has also led to the demand for more understanding of how H_2_S production is regulated as well as which signal transduction pathways mediate responses to this novel gasotransmitter.

Within the vascular wall, H_2_S is primarily produced by the enzyme cystathionine gamma-lyase (CSE) [9]. In agreement with other studies, our laboratory has demonstrated that CSE is expressed within mesenteric ECs and additionally, contributes to acetylcholine (ACh)-induced vasodilatory pathways under normal conditions in small resistance-size mesenteric arteries [11,12]. In addition, disruption of the endothelium abolishes dilatory responses to H_2_S donors, demonstrating that H_2_S vasodilatory signaling acts on ECs in an autocrine manner [12]. Multiple studies suggest H_2_S mediates endothelial-derived hyperpolarization (EDH)-like effects through EC intermediate conductance (IK_Ca_), small conductance (SK_Ca_), and large conductance (BK_Ca_) potassium channels, and through the transient receptor potential cation channel subfamily V member 4 (TRPV4) as well as smooth muscle cell ATP-sensitive potassium (K_ATP_) channels [13,14,15,16]. In mesenteric arteries specifically, we have shown that H_2_S dilatory responses are mediated through EC TRPV4 and BK_Ca_ channel activity [12,17]; however, it is unclear how this pathway is regulated.

Membrane cholesterol has been shown to regulate ion channel activity, mobility, membrane integration, and docking [18,19,20,21,22]. Membrane cholesterol has also been shown to negatively regulate both TRPV4 and BK_Ca_ channel activity and mobility of TRPV4 within the PM [18,23]. Considering that cholesterol regulates both TRPV4 and BK_Ca_, we postulated that EC PM cholesterol regulates H_2_S-mediated vasodilation in mesenteric arteries.

Since large and small arteries were observed to have differences in sensitivity to H_2_S donors, we postulated that an innate difference in EC PM cholesterol content of large and small mesenteric arteries could account for differences in sensitivity to H_2_S. We hypothesized that EC PM cholesterol is higher in large mesenteric arteries and limits H_2_S-mediated vasodilation.

## 2. Materials and Methods

### 2.1. Animals

All animal protocols were reviewed and approved by the Institutional Animal Care and Use Committee of the University of New Mexico School of Medicine and conform to the National Institutes of Health guidelines for animal use. Adult male Sprague Dawley rats (200–250 g) or adult female BL/6 J mice (20–25 g) were used in all experiments. Animals were housed in a pathogen-free animal care facility and maintained on a 12:12-h light-dark cycle. Animals were housed in polyacrylic cages and supplied with bedding and polycarbonate rodent tunnels. Animals were anesthetized with sodium pentobarbital (200 mg/kg, ip). While under deep anesthesia, a thoracotomy was performed and the left ventricle injected with heparin (100 units, 1 mL total), and animals immediately euthanized by exsanguination.

### 2.2. Isolation and Preparation of Artery Segments

Mesenteric cascades were isolated and bathed in chilled HEPES buffered physiological saline (HPSS) (mmol/L: 1.4 CaCl_2_, 0.03 EDTA, 10.10 glucose, 10 HEPES, 6 KCl, 2.13 MgSO_4_ and 134 NaCl) in a Petri-dish and cleaned of adipose tissue. Second- or third-order (large, lumen diameter > 300 µm) and sixth- or seventh-order (small, lumen diameter < 130 µm) artery segments were dissected and transferred to a single-vessel chamber (CH-1, Living Systems). Proximal ends of arteries were mounted on an inflow glass cannula, secured using silk ligature, and gently flushed to remove blood from the lumen. After securing the distal end to outflow glass cannula, arteries were pressurized to 70–75 mmHg with HPSS using a pressure servo controller (Living Systems) and superfused with HPSS at a rate of 5 mL/min at 37 °C. Prior to start of experiments, arteries were equilibrated in HPSS for 15 min and viability assessed using 100 mM KCl in the superfusate; arteries that did not constrict to KCl were discarded.

### 2.3. Vasodilation Studies (Artery Segments)

EC PM sterol manipulation was accomplished following a previously published protocol from our group with slight modification [24]. Following 10 min of washout of KCl, arteries were luminally perfused and incubated with vehicle (HPSS), MBCD (100 µM), MBCD + cholesterol (C495130, Sigma-Aldrich, St. Louis, MI, USA, 28 µM) or MBCD + epicholesterol (C6730-000, Steraloids, Newport, RI, USA, 20 µM) to evaluate H_2_S-mediated vasodilation following EC PM sterol manipulation. Other arteries were luminally perfused with vehicle (HPSS), MBCD + GSK 2193874 (5106, Tocris, Bristol, UK, 300 nM) or MBCD + iberiotoxin (ab120379, Abcam, Cambridge, UK, 100 nM) to evaluate TRPV4 or BK_Ca_ contribution to H_2_S-mediated vasodilation following EC PM sterol manipulation. Following incubation, arteries were constricted to ~50% resting diameter with the thromboxane mimetic, U46619 (~10^−8^ to 10^−9^ M for small arteries; ~10^−5^ to 10^−6^ M for large arteries) (Item No. 16450, Caymen Chemical, Ann Arbor, MI, USA). Artery lumen diameter following exposure to bolus concentrations of the H_2_S donor, NaHS (1, 10 or 100 µM, Alfa Aesar, Haverhill, MA, USA) was recorded using edge-detection software (IonOptix). Vasodilation is expressed as:$$\%\;\mathrm{Dilation}=\frac{\;\mathrm{NaHS}\;\mathrm{diameter}\;-\;U46619\;\mathrm{diameter}}{\mathrm{Calcium}\;\mathrm{Free}\;\mathrm{diameter}\;-\;U46619\;\mathrm{diameter}}\times 100$$

At the end of each study, arteries were incubated with calcium-free buffer to induce maximal dilation and calculate % active dilation. Only arteries without spontaneous vasomotion were included in the analysis.

### 2.4. Isolation of Artery Cascade for Immunofluorescence

Mesenteric cascades were isolated and bathed in chilled HPSS in a Petri-dish. The superior mesenteric artery (SMA) connected to three to four intact artery cascades were cleaned of adipose tissue and transferred to a single vessel chamber (CH-1, Living Systems). The proximal end of SMA was mounted on the inflow glass cannula and secured with silk ligature. The artery cascade was gently flushed with phosphate buffered saline (PBS) to remove blood from the lumen. The artery cascade was superfused with HPSS at a rate of 5 mL/min at RT. Following removal of blood from the lumen, artery cascades were fixed luminally with perfusion of 4% paraformaldehyde for 15 min followed by a 10 min flush with PBS.

### 2.5. Cholesterol Staining with Filipin III (Artery Cascades)

After fixation, artery cascades were luminally perfused and incubated with cholesterol marker filipin III (F4767, Sigma-Aldrich, St. Louis, MI, USA, 20 µg/mL) and EC glycocalyx marker DyLight 594 conjugated *Lycopersicon esculentum* (Tomato) lectin (DL-1177, Vector Laboratories, Burlingame, CA, USA, 20 µg/mL) for 30 min followed by 3 min incubation with nucleic acid stain SYTOX Green (S7020, Life Technologies, Carlsbad, CA, USA, 1:15,000) and 10 min flush with PBS. Artery cascades were then cut longitudinally and mounted *en face* using Aqua-Poly/Mount (Item No. 1860620, Polysciences Inc., Warrington, PA, USA.). EC PM cholesterol was assessed by acquiring z-stack images using a confocal microscope (TCS SP8, Leica Microsystems, Wetzlar, Germany) with a 63X objective. Confocal images were analyzed using Leica Application Suite X (LAS X, Leica Microsystems, Wetzlar, Germany) to determine the average filipin fluorescent peak intensity that colocalized with the tomato lectin signal. To determine lumen diameter, artery cascades were imaged using a 4X objective, and diameter was calculated by dividing artery circumference by π. Arteries were categorized as small if the lumen diameter was <199 µm and large if the lumen diameter was >200 µm. Endothelial cholesterol signal represents an average of the cholesterol signal in 20–30 cells/small artery and 45 cells/large artery.

### 2.6. Immunostaining for TRPV4 and BK_Ca_ Expression (Artery Cascades)

After fixation, artery cascades were luminally perfused and incubated with EC glycocalyx marker DyLight 594 conjugated as above at RT for 30 min followed by 10 min of washing. Artery cascades were then frozen in O.C.T. (Tissue Tek), cut in 10 µm cross-sections, and mounted on slides. Slides were air-dried at 37 °C for 30 min. Following 1-min wash in deionized water (DI H_2_O) and 1-min wash in PBS, slides were permeabilized with 0.1% Triton X-100 in PBS for 10 min and blocked with 5% donkey serum + 0.1% Triton X-100 in PBS for 1 h, all at RT. Slides were then stained overnight at 4 °C with primary antibodies against TRPV4 (Item No. 39260, Abcam, Cambridge, UK, 1:100) or BK_Ca_ (75-022, Antibodies Inc., Davis, CA, USA, 1:100) diluted in 0.1% gelatin in PBS. Slides were incubated for 1 h at RT with DyLight 650 conjugated donkey anti-rabbit and Alexa Fluor 647 conjugated donkey anti-mouse secondary antibodies to detect TRPV4 and BK_Ca,_ respectively. Nuclei were stained with SYTOX Green (S7020, Life Technologies, Carlsbad, CA, USA, 1:15,000) for 10 min at RT. Slides were mounted using Aqua-Poly/Mount (Item No. 1860620, Polysciences Inc., Warrington, PA, USA). TRPV4 or BK_Ca_ signal in the EC PM was assessed by acquiring z-stack images as described above and analyzed using Leica Application Suite X (LAS X, Leica Microsystems, Wetzlar, Germany) to determine the average TRPV4 or BK_Ca_ antibody fluorescent intensity that colocalized with the tomato lectin signal.

### 2.7. Proximity Ligation Assays (Artery Cascades)

EC PM TRPV4 and BK_Ca_ interactions were evaluated using Duolink proximity ligation assay (PLA) kit (DUO92101, Sigma). Before fixation, artery cascades were perfused and incubated with vehicle (HPSS), MBCD (100 µM), MBCD + cholesterol (28 µM), or MBCD + epicholesterol (20 µM) for 30 min at 32 °C to assess TRPV4 and BK_Ca_ association under basal and sterol manipulated conditions. After fixation, artery cascades were perfused and incubated with EC glycocalyx marker as described above. Artery cascades were then frozen, cut, and permeabilized as described. Following the Triton X-100 permeabilization, slides were blocked with Duolink (DUO94003, Sigma) blocking buffer for 1 h at 37 °C. Slides were stained overnight at 4 °C with primary antibodies against TRPV4 (1:100) and BK_Ca_ (1:100) diluted in Duolink antibody diluent. Negative controls were incubated in buffer lacking both primary antibodies or including only one of the two primary antibodies. Slides were incubated for 1 h at 37 °C with Duolink PLUS and MINUS probes at a 1:5 dilution followed by amplification with *Duolink* In Situ *Reagent Red* for 100 min at 37 °C. Nuclei were stained with SYTOX Green (S7020, Life Technologies, Carlsbad, CA, USA, 1:15,000) for 10 min at RT then mounted using Duolink mounting media. Association of TRPV4 and BK_Ca_ was assessed by acquiring z-stack images using a confocal microscope (TCS SP5, Leica Microsystems, Wetzlar, Germany) with 63X objective. Confocal images were analyzed using ImageJ to quantify the number of PLA dots corresponding with the tomato lectin signal to determine proximity of targets within the EC. Data are expressed as the number of PLA puncta/Total area of selected ROI.

### 2.8. Experimental Design and Data Analysis

Data are presented as means ± SD and were analyzed using unpaired t-test or two-way ANOVA as appropriate (GraphPad Prism). Differences detected by ANOVA were compared with Sidak’s multiple comparisons post hoc test. *p* < 0.05 was considered statistically significant for all analyses. Power analysis of the data in the dilation study of arteries treated with cholesterol or epicholesterol indicated that the sample sizes of 3 to 5 replicates had a power of 0.78 for an α of 0.05.

## 3. Results

### 3.1. Cholesterol Reduction Augments H_2_S-Mediated Vasodilation in Large but Not Small Mesenteric Arteries

Previous findings demonstrate that a single vascular bed can have regional differences in CSE expression [25,26] and current studies examined potential differences in vasodilatory responses to H_2_S. Studies examining the dilatory functions of resistance size arteries used arteries between 60–180 µm inner lumen diameter where inhibition of CSE has been shown to attenuate ACh-induced vasodilation in small mesenteric arteries [11]. Unlike small arteries, CSE inhibition with beta cyanoalanine (BCA, 100 µM) had no effect on ACh-induced vasodilation (10 µM) in large arteries > 300 µm compared to vehicle conditions (87.8 ± 25.6 vs. 90.3 ± 11.9%). A range of ACh concentrations were tested showing the same results. Previous reports have also observed differences in sensitivity in response to H_2_S donors between large and small arteries [16]. That is, small arteries dilated to 1 µM H_2_S, whereas large arteries did not. This suggests there may be regional innate regulators of H_2_S signaling that mediate the difference in sensitivity seen in large and small arteries.

Since cholesterol has been shown to inhibit BK_Ca_ (reviewed in [19]) and TRPV4 activity [18], the effects of cholesterol removal from EC PM on artery dilatory responses to H_2_S donor, NaHS were examined. We previously showed that ECs are required for H_2_S-induced vasodilation, since endothelium-denuded arteries failed to dilate in response to H_2_S donors [12]. To target EC PM, artery segments were perfused luminally with MBCD at 100 µM, a concentration shown to reduce PM cholesterol without affecting caveolae structures [23]. Reduction of EC PM cholesterol uncovered NaHS-induced vasodilation at both 10 and 100 µM NaHS in large arteries (Figure 1A) but had no effect on dilation in small arteries (Figure 1B). Small arteries were more sensitive and exhibited greater dilation to NaHS (even at 1 µM), confirming our previous observation that small arteries are more sensitive to H_2_S. Additionally, we observed that this difference in sensitivity to H_2_S between large and small arteries occurs in mice. Similar to rat arteries, H_2_S induced robust dilation in mouse small mesenteric arteries with minimal to no dilation in large arteries (Figure 1C). Although we did not examine the effect of cholesterol depletion in the mouse arteries to verify that the mechanism is the same, the similarity of the responses in large and small arteries suggest differences in the sensitivity to H_2_S are also present in other species.

### 3.2. EC PM Cholesterol Content Is Higher in Large Mesenteric Arteries

As cholesterol removal increased H_2_S-induced vasodilation in large but not small arteries, EC PM cholesterol content was evaluated in large and small arteries. Analysis of en face fluorescence of arteries stained with the cholesterol marker, filipin III (20 µg/mL) in EC labeled with the glycocalyx marker tomato lectin (20 µg/mL) assessed EC PM cholesterol content (Figure 2A) and showed that large artery ECs contain higher PM cholesterol content compared to small artery ECs (large 145.20 ± 13.67, small: 59.85 ± 5.50 A.U., *p* = 0.0004, *n* = 5 animals/group) (Figure 2B). These data demonstrate that ECs in large arteries contain higher cholesterol levels and support the hypothesis that higher EC PM cholesterol content might dampen responses to H_2_S in large arteries. This suggests that adding cholesterol into EC PM of small arteries should reduce small artery sensitivity to H_2_S-induced dilation.

### 3.3. EC PM Cholesterol Supplementation Abolishes H_2_S-Mediated Vasodilation in Small Arteries

MBCD molecules chelate sterols by forming an encapsulated sterol structure. This also makes them an excellent vehicle for sterol delivery [22]. The addition of cholesterol into EC PM was achieved by luminal perfusion with MBCD + cholesterol (1:5) [24] to target EC specifically. Supplementation of cholesterol into EC PM of small arteries abolished NaHS-induced vasodilation (Figure 3). To dissect whether the effect of cholesterol is through differences in sterol content of the PM, dilatory responses were also assessed after supplementation with the cholesterol enantiomer, epicholesterol. Cholesterol and epicholesterol differ in the spatial orientation of a hydroxyl group on the position 3 chiral center [27]. This causes cholesterol and epicholesterol to have different orientations and protein binding patterns within lipid membranes; however, there are minimal differences in the physical properties of epicholesterol- and cholesterol-enriched membranes [16]. The main difference is therefore that only cholesterol has specific interactions with lipid-embedded proteins. Epicholesterol had a modest effect to diminish dilation but did not affect responses as profoundly as supplementation with cholesterol, suggesting the effects of cholesterol are through at least in part through direct binding and inhibition of channel activity but may also be in part through changes in PM fluidity and structure. Future studies examining endothelial PM structure after cholesterol supplementation could address these effects.

### 3.4. Effect of EC PM Cholesterol Reduction to Uncover H_2_S-Mediated Vasodilation in Large Arteries Is through EC TRPV4 and BK_Ca_ Channels

Since cholesterol reduction increased H_2_S-induced vasodilation in large arteries, we then evaluated whether this dilation was mediated through TRPV4 and BK_Ca_ channels. TRPV4 and BK_Ca_ expression, evaluated via immunofluorescence, demonstrated that large and small arteries express both channels (Figure 4A).

Large arteries were perfused luminally with MBCD to reduce EC PM cholesterol in the presence of the BK_Ca_ inhibitor iberiotoxin or TRPV4 inhibitor GSK, and responses to 10 µM NaHS were recorded. Both iberiotoxin and GSK abolished the dilation to H_2_S after cholesterol reduction (Figure 4B–E). This demonstrates that the H_2_S dilation uncovered by endothelial cholesterol reduction in large arteries is mediated through TRPV4 and BK_Ca_ channels. Additionally, the data also demonstrate that it is not a lack of TRPV4 or BK_Ca_, but rather higher PM cholesterol content appears to inhibit ion channel activation.

### 3.5. TRPV4 and BK_Ca_ Association in Large and Small Arteries–PLA Studies

As there was expression of both TRPV4 and BK_Ca_ in large and small arteries, the effects of PM cholesterol to limit H_2_S signal transduction through limiting association of the two ion channels was then assessed. There was no difference in PLA puncta count/total area between large and small arteries (Figure 5A–C). Negative controls (Figure 5D) show that no puncta were observed when either of the antibodies were omitted. Neither cholesterol reduction, cholesterol augmentation nor epicholesterol treatment had an effect. This suggests the effect of cholesterol to limit H_2_S-mediated signaling through TRPV4 and BK is through direct inhibition of channel activity rather than changes in channel association and mobility. Future studies will address this hypothesis.

## 4. Discussion

Resistance-sized arteries (<180 μm) are critical modulators of blood pressure and flow. Nitric oxide signaling has been shown to profoundly affect large conduit vessels, while in resistance-size arteries, EDH effects such as that exerted by H_2_S are more apparent. We observed that loss of endogenous H_2_S attenuated endothelial-dependent vasodilation in small but not large arteries. This demonstrates innate differences between large and small arteries in one or more components of the H_2_S vasodilator pathway. As membrane cholesterol can negatively regulate BK_Ca_/TRPV4 activity and mobility in the PM, the present study addressed the hypothesis that EC PM cholesterol limits H_2_S-mediated vasodilation in large arteries. Exploiting the natural differences in large and small arteries, we found; (1) large arteries are less sensitive to H_2_S donor NaHS, but become sensitive to H_2_S donors after EC PM cholesterol reduction; and this higher sensitivity to H_2_S seen in small arteries is not unique to rats but is also present in mice; (2) under normal basal conditions, EC of large mesenteric arteries contain higher PM cholesterol content; (3) small arteries are more sensitive to H_2_S, and cholesterol supplementation into EC PM of small arteries abolishes vasodilation to H_2_S donors with no effect of the cholesterol enantiomer epicholesterol; (4) both large and small arteries express TRPV4 and BK_Ca_ channels; however; (5) this H_2_S-mediated TRPV4/BK_Ca_ vasodilatory pathway is inactive in large arteries and only uncovered after EC PM cholesterol reduction and; (6) sterol manipulation had no effects on TRPV4 and BK_Ca_ association in EC. Collectively, these data demonstrate EC PM cholesterol limits H_2_S-mediated vasodilation.

A potential mechanism by which PM cholesterol can limit H_2_S-mediated vasodilation is through direct binding and inhibition of TRPV4/BK_Ca_ channel activity [20,21]. Epicholesterol supplementation did not affect H_2_S-induced vasodilation in small arteries, suggesting that the mechanism by which PM cholesterol influences H_2_S signaling is not due to alterations in PM fluidity, but rather via direct interaction with the TRPV4 and/or BK_Ca_ channel.

Another potential mechanism by which EC PM cholesterol can limit H_2_S-mediated vasodilation is through facilitating proximity of TRPV4/BK_Ca_ channels and spatial distribution of the channels along the PM, sorting the channels into signaling microdomains. The lack of effect of epicholesterol supplementation on H_2_S-induced vasodilation in small arteries does not fully negate this possibility. Structurally, cholesterol and epicholesterol differ in the spatial orientation of a hydroxyl group on the position 3 chiral center [27]. Though they have different orientations within lipid membranes, there are minimal differences in the physical properties of epicholesterol- and cholesterol-enriched membranes. The main difference between the two enantiomers is their specific interaction with lipid-embedded proteins. There is abundant evidence showing that cholesterol binds and regulates the activity of multiple ion channels, including BK_Ca_ and TRPV4. Additionally, cholesterol can affect the integration of ion channels and other proteins into the PM [28]. This provides the rationale to employ epicholesterol to delineate direct binding from alterations in membrane structure and fluidity. Furthermore, recent findings demonstrate that TRPV4 localization and signaling in myoendothelial projections are important for vasodilatory responses [29]. TRPV4 has been shown to have a cholesterol-binding site and, when cholesterol-bound, the ion channel has decreased mobility within the PM [18]. Hence, it is possible that cholesterol might affect the distribution and activity of TRPV4 or BK_Ca_ within PM microdomains. However, epicholesterol had no effect on H_2_S-mediated vasodilation, and sterol manipulation did not affect the association of the two ion channels, suggesting the effect of cholesterol is to bind to and inhibit channel activity.

There is controversy over the expression and activity of BK_Ca_ channels on EC. In this study, TRPV4 and BK_Ca_ inhibitors were perfused luminally in large mesenteric arteries to target the ion channels expressed in EC. The pressure myography data (Figure 4) demonstrates that H_2_S-mediated vasodilation apparent in large arteries after cholesterol reduction is dependent on TRPV4 and BK_Ca_ channels expressed on the ECs. This agrees with our previous findings that inhibitors of BK_Ca_ employed luminally have a more profound effect on H_2_S-mediated vasodilation than when applied abluminally [12] and it also agrees with electrophysiological characterization of BK channels in ECs.

Previous reports demonstrate that in response to hypoxia, the systemic vasculature exhibits decreased EC PM cholesterol and increased endothelial-dependent hyperpolarization, which is mediated through TRPV4 and BK_Ca_ [17,23]. In agreement, the current results demonstrate that a loss of PM cholesterol uncovers H_2_S signaling in EC mediated through TRPV4 and BK_Ca_. The previous studies demonstrate that exposure (acute: 48 h or chronic: 4 weeks) to hypoxia decreases EC PM cholesterol. The current data further demonstrate greater cholesterol content in EC PM of large arteries suggesting that either acute manipulation or chronic cholesterol depletion may similarly uncover an H_2_S sensitive vasodilatory pathway mediated through TRPV4 and BK_Ca_ in large arteries. These findings are also in agreement with other reports showing innate endothelial differences in the H_2_S axis, driven by differences in flow and shear stress, namely CSE expression and activity varies along a single vascular bed [25,26].

## 5. Conclusions

In conclusion, the present work describes native regional differences in EC PM cholesterol within a single vascular bed, contributing to a difference in sensitivity to H_2_S-mediated vasodilation through TRPV4 and BK_Ca_ channel activation. This study demonstrates a novel role of EC PM cholesterol to limit H_2_S-mediated vasodilation in mesenteric arteries through EC TRPV4 and BK channels. These findings define native regional differences of EC PM cholesterol content which appear to account for the differences in H_2_S sensitivity between large and small mesenteric arteries, and have the potential to define mechanisms leading to regional differences in EC function and pathophysiological susceptibility.

## Figures and Tables

**Figure 1 antioxidants-11-01680-f001:**
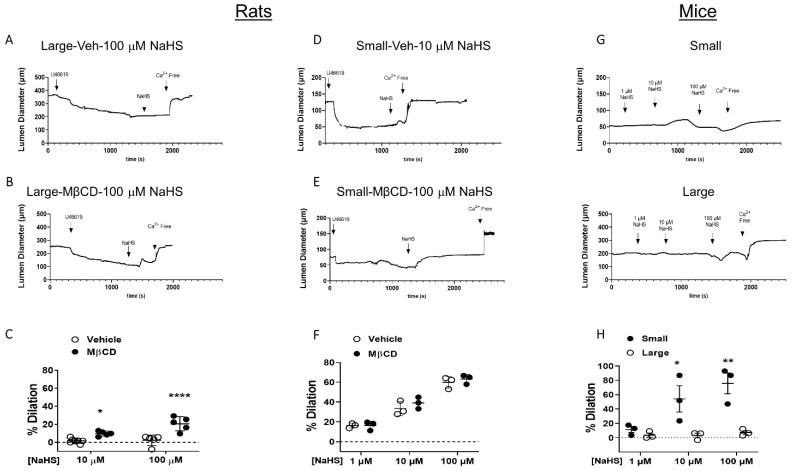
H_2_S-induced vasodilation in rat and mouse mesenteric arteries. Representative traces and data summary of H_2_S-induced vasodilation following cholesterol reduction with MβCD (100 μM) in rat large arteries (**A**–**C**). Mean ± S.D. * *p* = 0.026 vs. vehicle; **** *p* < 0.0001 vs. vehicle and rat small arteries (**D**–**F**). Mean ± S.D. (**G**–**H**). H_2_S-induced vasodilation in mouse large and small mesenteric arteries. Mean ± S.D. ** p* = 0.01, *** p* = 0.0011 vs. large, two-way ANOVA with Sidak’s multiple comparison test.

**Figure 2 antioxidants-11-01680-f002:**
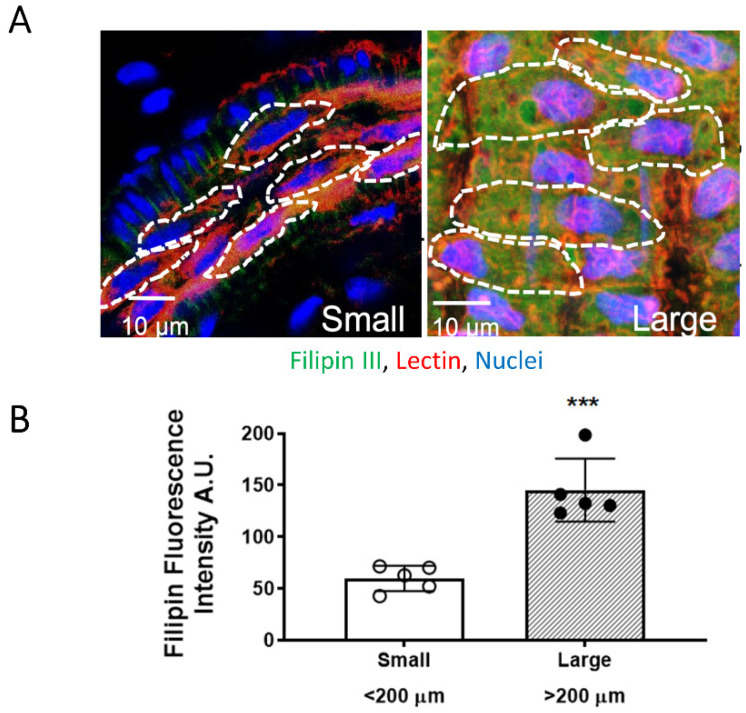
Cholesterol content in large (>200 μm) and small (<200 μm) mesenteric arteries. (**A**). Representative en face images of cholesterol marker, filipin III (green, 20 μg/mL) and endothelial glycoprotein marker Lectin (red, 20 μg/mL) acquired at 63x with additional digital zoom of small artery for visualization of cells. Nuclei are labeled with SYTOX (blue). (**B**). Mean ± S.D. *** *p* = 0.0004, unpaired *t* test.

**Figure 3 antioxidants-11-01680-f003:**
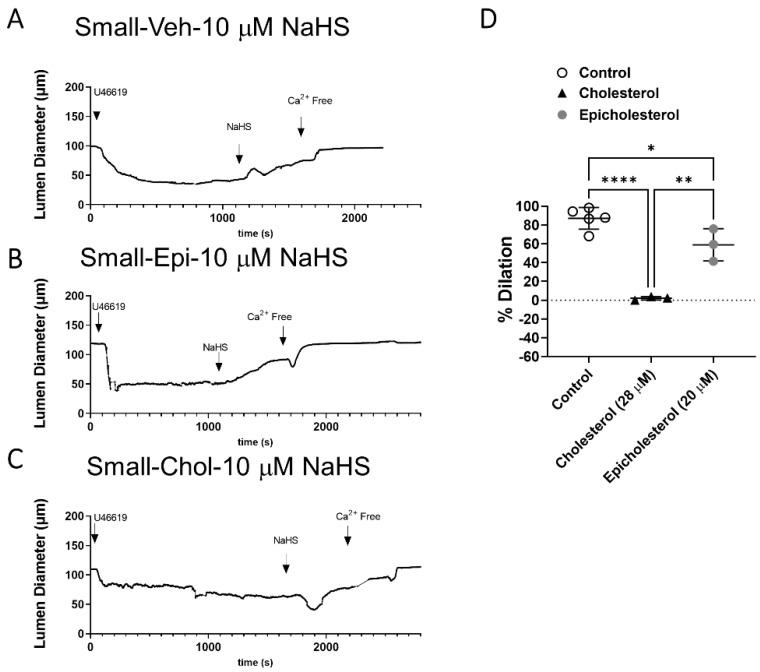
Representative traces and data summary of H_2_S-induced vasodilation under (**A**). vehicle (Veh) condition or following (**B**). epicholesterol (Epi, 20 μM) or (**C**). cholesterol (Chol, 28 μM) supplementation into EC PM of small mesenteric arteries. Summary data of H_2_S dilation in cholesterol and epicholesterol treated small arteries (**D**) Mean ± S.D. * *p* = 0.04, ** *p* = 0.0012, **** *p* < 0.001, one-way ANOVA with Sidak’s multiple comparison test.

**Figure 4 antioxidants-11-01680-f004:**
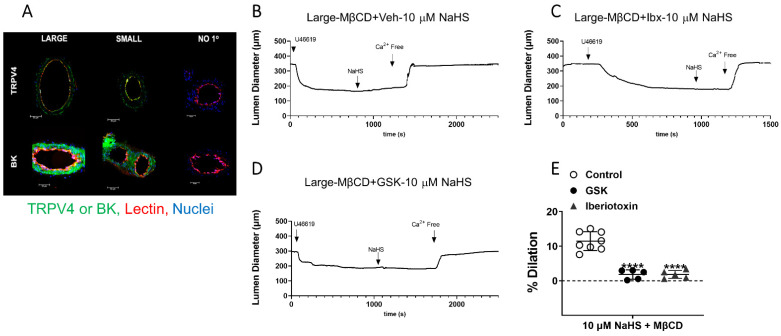
H_2_S-induced vasodilation following cholesterol reduction in large arteries is mediated by activation of TRPV4 and BK channels. (**A**). Representative images of EC PM TRPV4 and BK (green), endothelial glycoprotein marker Lectin (red) in cross-sections of large and small mesenteric arteries. Nuclei are labeled with SYTOX (blue). “NO 1°” shows the no primary Ab negative controls. Representative traces and data summary of H_2_S-mediated vasodilation following cholesterol reduction (**B**) in large arteries in the presence of BK inhibitor iberiotoxin ((**C**), Ibtx, 100 nM) or TRPV4 inhibitor GSK 2193874 ((**D**), GSK, 300 nM) Summary data of H_2_S-induced dilation in cholesterol depleted arteries (**E**). Mean ± S.D. **** *p* < 0.0001 vs. control, one-way ANOVA with Sidak’s multiple comparison test.

**Figure 5 antioxidants-11-01680-f005:**
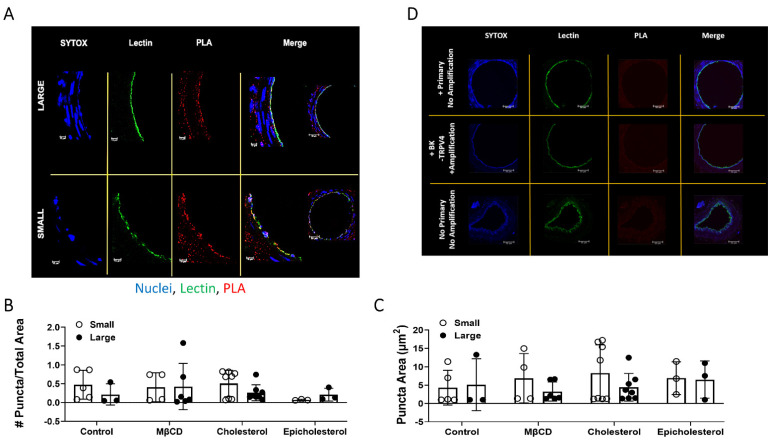
Endothelial TRPV4 and BK association in large and small mesenteric arteries following sterol manipulation. (**A**). Representative images of mesenteric artery cross-sections with endothelial glycoprotein marker Lectin (green) and PLA puncta for TRPV4/BK association (red). Nuclei are labeled with SYTOX (blue). (**B**). Number (#) of puncta count /total area and (**C**). area of puncta from 23 arteries/size group per animal. Mean ± S.D. *n* = 4 animals/group. (**D**). Primary antibodies without ligation or amplification (+Prim/No Amp); inclusion of BK but not TRPV4 primary antibody with ligation and amplification (+BK/-TRPV4/+Amp); and No primary No amplification as negative controls.

## Data Availability

All of the data is contained within the article.

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
