# Peer review of "Role of Cholesterol in the Regulation of Hydrogen Sulfide Signaling within the Vascular Endothelium"

_antioxidants, 2022, doi:10.3390/antiox11091680_

Round 1
Reviewer 1 Report
In this manuscript, Mendiola and colleagues identify a new mechanisms whereby membrane cholesterol content determines endothelium-dependent vasodilation to H2S. Previous studies by this group show that H2S is a crucial mediator of endothelium-dependent vasodilation. Findings on cholesterol-dependence improve our understanding of H2S-induced signaling. Authors show that lowering cholesterol content in endothelial membrane can enhance dilation to H2S whereas increasing cholesterol content can attenuate dilation.
1. Methods state that U46619 was used at nano molar concentration for constricting small arteries and at 1-10 M in large arteries. Please check if this is accurate as the difference in concentration is striking.
2. Line 188 needs a citation for the statement that cholesterol inhibits TRPV4 and BK channel activity.
3. Figure 1- show traces in small arteries for 10 microM NaHS without MBCD. Does MBCD have a similar effect on NaHS dilation of mouse arteries?
4. Figure 3 control experiment- authors can use cholesterol imaging, as done in figure 2, to show that MBCD + cholesterol approach increases EC membrane cholesterol.
5. Figure 3 legend: subfigure labels (A-D) are missing. Also, uM should be replaced by micro (symbol)M.
6. Figure 4- uM should be replaced by micro (symbol)M.
7. Line 280- I suggest removing this sentence as there is no evidence for effects on channel activity.
8. Figure 5- PLA punch are difficult to see. Please increase the intensity/contrast.
Author Response
Thank you for the helpful review. We have addressed your queries in the attached file.
NLK

Reviewer 2 Report
The authors performed several lines of experimentation suggesting that the higher cholesterol content in large blood vessels compared to small blood vessels is responsible for the lack of effects of H2S on vessel dilation in the large vessels. Cholesterol is known to inhibit BKCa and TRPV4 channels required for vessel dilation. The experiments are well-designed and the data analysis appears appropriate. I have only a few concerns as shown below.
Major comments:
Comment 1: Results Figure 1: Please explain why the calcium free-medium increases the vessel lumen diameter.
Comment 2: In Figure 3, why were not equal concentrations of cholesterol (28 uM) and epicholesterol (20 uM) added? Perhaps a significant effect of epicholesterol would be observed at 28 uM instead of 20 uM. Due to the spread in the epicholesterol data more data points are needed as well.
Comment 3: The resolution of the figures should be increased slightly to help read the small font. This is most apparent in Figure 1G.
Comment 4: A biochemical or mass spectrometry assay of small or large vessel cholesterol content would be a nice confirmation of the filipin III fluorescent measurements. Does filipin III have access to all membranes in the vessels or only membranes on the luminal side? Line 114 states “artery cascades were luminally perfused and incubated with cholesterol marker filipin III”.
Comment 5: The reason for choosing to submit the manuscript to Antioxidants is not apparent due to the lack of discussion of biological oxidation.
Minor comments:
Abstract p. 1 line 17: the beta symbol in MBCD (methyl—cyclodextrin) did not show up in the pdf document.
p. 3 line 98: the mu in µM did not appear in the pdf document.
p. 104: Superior -> The superior
p. 4 line 106: to -> to a
p. 4 line 109: Artery -> The artery
p. 4 line 110: lumen -> the lumen
p. 4 line 11: by -> by a
p. 4 line 116: Please italicize Lycopersicon esculentum
p. 4 line 125: using -> using a
p. 4 line 125:
p. 4 line 185: to -> in response to
Figure 1B and 1E panel titles and Figure 4B, 4C, and 4D panel titles: Please change the B to a beta in MBCD.
Figure 1 C Y-axis title: Dllation -> Dilation
p. 6 lines 225 and 226: 20μg/ml -> 20 μg/ml [insert space]
p. 6 line 231: it -> them
p. 6 line 246: cholesterol -> cholesterol (Chol)
p. 6 line 246: epicholesterol -> epicholesterol (Epi)
p. 7 line 265: eBK -> BK
p. 7 line 265: green -> (green)
p. 8 line 285: in red -> (red)
p. 8 line 305: H2S -> H2S,
p. 8 line 316: with -> with the
p. 9 line 345: and -> and it also agrees with
p. 10 line 379: Please alter or remove the Data availability Statement.
Author Response
Thank you for the helpful critique. We have revised the manuscript and addressed your queries as outlined in the attached file.
NLK

Round 2
Reviewer 1 Report
No further comments.